# Direct Oral Anticoagulants for Stroke Prevention in Special Populations: Beyond the Clinical Trials

**DOI:** 10.3390/biomedicines11010131

**Published:** 2023-01-04

**Authors:** Andreina Carbone, Roberta Bottino, Antonello D’Andrea, Vincenzo Russo

**Affiliations:** 1Department of Cardiology, University of Campania “Luigi Vanvitelli”, 80131 Naples, Italy; 2Unit of Cardiology and Intensive Coronary Care, “Umberto I” Hospital, 84014 Nocera Inferiore, Italy; 3Monaldi Hospital, P.zzale Ettore Ruggeri, 80131 Naples, Italy

**Keywords:** direct oral anticoagulant, stroke prevention, atrial fibrillation, elderly, frail, malignancy, extreme weight, bioprosthetic valve

## Abstract

Currently, direct oral anticoagulants (DOACs) are the first-line anticoagulant strategy in patients with non-valvular atrial fibrillation (NVAF). They are characterized by a more favorable pharmacological profile than warfarin, having demonstrated equal efficacy in stroke prevention and greater safety in terms of intracranial bleeding. The study population in the randomized trials of DOACs was highly selected, so the results of these trials cannot be extended to specific populations such as obese, elderly, frail, and cancer patients, which, on the other hand, are sub-populations widely represented in clinical practice. Furthermore, due to the negative results of DOAC administration in patients with mechanical heart valves, the available evidence in subjects with biological heart valves is still few and often controversial. We sought to review the available literature on the efficacy and safety of DOACs in elderly, obese, underweight, frail, cancer patients, and in patients with bioprosthetic heart valves with NVAF to clarify the best anticoagulant strategy in these special and poorly studied subpopulations.

## 1. Introduction

Atrial fibrillation (AF) is the most common arrhythmia in clinical practice, and it is associated with an increased risk of ischemic stroke and mortality [1]. The estimated prevalence of AF in adults is between 2% and 4%, increasing in view of the progressive increase in life expectancy [1].

The European Society of Cardiology (ESC) guidelines recommend the assessment of stroke risk using the CHA2DS2-VASc score, considering oral anticoagulation (OAC) prescription for scores of ≥1 in males and ≥2 in females [1]. OAC with vitamin K antagonists (VKAs) decrease the risk of stroke by 68% [2], but at the cost of routine monitoring of anticoagulation levels [International Normalized Ratio (INR) determination], due to their pharmacokinetic variability and frequent food and drug interactions [2,3].

Direct oral anticoagulants (DOACs) have emerged as an alternative to VKAs as they have demonstrated comparable efficacy in stroke prevention with less major bleeding than warfarin in patients with non-valvular AF [4]. Moreover, DOACs do not require routine monitoring of anticoagulation parameters as they are characterized by a predictable pharmacokinetic and few drug and food interactions [4]. Dabigatran etexilate is the first established factor IIa (thrombin) inhibitor. It is a prodrug converted into the active form of dabigatran by microsomal carboxylesterases in the liver [1,3]. The mechanism of action of rivaroxaban, apixaban, and edoxaban is the inhibition of prothrombinase complex-bound and clot-associated factor Xa, resulting in a reduction of the thrombin burst during the propagation phase of the coagulation cascade [1].

For special populations, such as elderly and frail patients, subjects with extreme body weight, cancer, and bioprosthetic heart valves (BHV), the indications and the choice of the optimal OAC therapy are a challenge because of the poor representation of these populations in major randomized clinical trials (RCTs). This review aims to summarize the current relevant literature regarding the use of DOACs in the special populations mentioned above.

## 2. DOACs in Elderly Patients

Older age increases both ischemic (especially stroke events) and bleeding risk [5,6]. In older patients with AF, VKAs prevent stroke with increased bleeding risk and the need for frequent INR monitoring [7,8]. Likewise, the elderly often present with multiple comorbidities, so an integrated approach to prevent stroke events, including tailored therapy and careful drug-dose monitoring, is mandatory [9,10,11]. To date, DOACs are the first-line therapy for stroke prevention and are characterized by a more favorable pharmacological profile than VKA. The efficacy and safety profile of DOACs in patients >75 years have been analyzed in various studies, but their use in octogenarians and frail patients is still poorly explored. Table 1 summarizes the main characteristics of principal DOAC trials and outcomes in patients ≥75 years.

In a sub-analysis of the RE-LY (Randomized Evaluation of Long-Term Anticoagulation Therapy) trial [12,13], both dabigatran 110 mg BID and 150 mg BID were found comparable to VKAs for the combined endpoint of stroke/systemic embolism and major bleeding. However, comparing dabigatran dosages with warfarin showed a lower risk of intracranial bleeding but similar or higher extracranial bleeding events [13]. These results were confirmed for stroke and systemic embolism events in both patients ≥80 years [dabigatran 110 mg bid (HR = 0.75; *p* = non-significant [NS]) and 150 mg bid (HR = 0.67; *p* = NS)] and ≥85 years old [dabigatran 110 mg bid (HR = 0.52, *p* = NS) and 150 mg bid (HR = 0.70; *p* = NS)] [14].

Finally, compared to warfarin, the use of dabigatran in women aged 75 years and older seems to be related to an increased risk of major gastrointestinal bleeding (HR 1.50; *p* < 0.05) as well as in men aged 85 years and older (HR 1.55; *p* < 0.05) [15]. In women ≥85 years, no effect on mortality was found using the dabigatran [15].

A sub-analysis of ROCKET-AF (Rivaroxaban Once-daily oral direct factor Xa inhibition Compared with vitamin K antagonism for Prevention of Stroke and Embolism Trial in Atrial Fibrillation) trial found that in elderly patients (≥75 years old; *n* = 6229), rivaroxaban had similar efficacy in reducing stroke and systemic embolism (HR = 0.88; *p* < 0.05) with a lower rate of intracranial bleeding (HR = 0.80; *p* < 0.05) when compared with warfarin [16]. Anyway, patients on rivaroxaban showed a higher risk of the combined bleeding endpoint due to more frequent non-major gastrointestinal bleeding (2.81% versus 1.66%/100 patient years; HR 0.70; *p*: 0.0002) [16].

The ARISTOTLE (Apixaban for Reduction in Stroke and Other Thromboembolic Events in Atrial Fibrillation) trial included 5678 patients ≥75 years (31%) and 2436 patients (18%) ≥80 years at baseline [17]. Apixaban showed absolute clinical benefits in the older population with a significant reduction of stroke or systemic embolism (HR 0.81; *p* < 0.05), major bleeding (HR: 0.66; *p* < 0.05), and intracranial hemorrhage (HR 0.36; *p* < 0.05) in patients ≥80 years compared to warfarin [17].

In a study cohort of 14,214 AF patients (mean age 78.1), the risk of stroke/systemic embolism (HR: 0.65, *p* < 0.001), major bleeding (HR: 0.53, *p* < 0.001), and gastro-intestinal bleeding (HR: 0.53, *p* < 0.001) was lower in the apixaban group when compared with the same number of elderly patients on warfarin (7107 patients in each group) [18]. These results were later confirmed by Yao et al. in a cohort of AF patients with a median age of 73 years old, in which apixaban patients showed a 33% lower risk of stroke/systemic embolism (HR = 0.67 *p* = 0.04) and 55% lower risk of major bleeding (HR 0.45, *p* < 0.001) compared to VKAs [19].

In the ENGAGE AF-TIMI 48 (Effective Anticoagulation with Factor Xa Next Generation in Atrial Fibrillation–Thrombolysis in Myocardial Infarction 48), 40.2% of the enrolled patients were aged over 75 years old and 17% over 80 years old (8474 and 1440 patients, respectively). While the incidence of stroke/systemic embolism in AF patients aged ≥75 years was similar between edoxaban and warfarin (1.5% per year in warfarin group; 1.18% per year in edoxaban group; HR vs. warfarin, HR 0.79; *p* < 0.001 for noninferiority, *p* = 0.02 for superiority), the incidence of major bleeding was lower in the edoxaban group (3.43% per year in warfarin group and 2.75% in edoxaban group) [20].

In a meta-analysis of the randomized controlled trial (RCT), DOACs were associated with equal or greater efficacy than VKAs, without relevant bleeding among patients of ≥75 years [21].

The ELDERCARE-AF (Edoxaban Low-Dose for EldeR CARE AF patients) compared the safety and efficacy of once-daily edoxaban 15 mg versus placebo in AF Japanese patients aged ≥80 years for whom standard oral anticoagulants were contraindicated [22,23]. Edoxaban 15 mg was superior to a placebo in reducing stroke or systemic embolism, but the study was limited by race clustering (all patients were Asian) and by a low mean of the body weight of the cohort (about 50 kg).

The safety profile of DOACs is also maintained in patients aged ≥90 years, as shown by the analysis of 15,756 AF patients sourced from the Taiwan National Health Insurance Research Database (NHIRD) [24]. Indeed, while the risk of ischemic stroke was found to be comparable between VKAs and DOAC users among elderly patients (4.07%/y versus 4.59%/y; HR: 1.16; *p* = 0.654), the DOAC group showed a lower risk of intracranial hemorrhage (0.42%/year versus 1.63%/year; HR 0.32; *p* = 0.044) [24].

In addition, it is worth mentioning that inappropriate DOAC dosage prescription affects up to 15% of AF patients [25], especially older patients [1,25].

In a multicenter study of AF patients aged ≥80 years who received DOAC treatment (*n* = 253), Carbone et al. showed that nearly one-third of octogenarians with AF received an inappropriate dose of DOACs [26]. Several clinical factors were associated with DOAC overdosing [diabetes mellitus type II (OR 18; *p* < 0.001), previous bleeding (OR 6.4; *p* = 0.03)] or underdosing [male gender (OR 3.15; *p* < 0.001), coronary artery disease (OR 3.60; *p* < 0.001), and higher body mass index (OR 1.27; *p* < 0.001)] [26]. Octogenarians with inappropriate DOAC underdosing showed less survival (*p* < 0.001) [26].

**Table 1 biomedicines-11-00131-t001:** Characteristics of DOAC principal trials and outcomes in patients aged ≥75 years.

	RE-LY [12]	ROCKET-AF [16]	ARISTOTLE [17]	ENGAGE [20]
DOACs vs. VKAs Dose Reduced dose	dabigatran 150 mg bid 110 mg bid	rivaroxaban 20 mg qd 15 mg qd	apixaban 5 mg bid 2.5 mg bid	edoxaban 60 mg qd 30 mg qd
Patients (n)	18,113	14,264	18,201	14,071
Age (mean in years)	72	73	70	72
Patients ≥75 years, n (%)	7258 (40)	6229 (44)	5678 (31)	5668 (40)
ClCr in ≥75 years at baseline	≥80 mL/min: 12% 50–79 mL/min: 57% <50 mL/min: 26%	Median 55 mL/min (IQR 44, 68).	>80 mL/min: 10.5% 51–80 mL/min: 51.5% 31–50 mL/min: 33.6% ≤30 mL/min: 3.9%	>80 mL/min: 12% 51–80 mL/min: 52% ≤50 mL/min: 37%
Stroke or systemic embolism in patients ≥75 years				
Event rates (DOACs vs. VKAs %/years) HR (or RR for dabigatran) (IC 95%)	1.9 (110 mg bid) vs. 2.1 1.4 (150 mg bid) vs. 2.1 0.88 (0.66–1.17) (110 bid) 0.67 (0.49–0.90) (150 bid)	2.3 vs. 2.9 0.80 (0.63–1.02)	1.6 vs. 2.2 0.71 (0.53–0.95)	1.9 vs. 2.3 0.83 (0.67–1.04)
Major bleedings in patients ≥75 years				
Event rates (DOACs vs. VKAs %/years) HR (or RR for dabigatran) (IC 95%)	4.4 (110 mg bid) vs. 4.4 5.1 (150 mg bid) vs. 4.4 1.01 (0.83–1.23) (110 bid) 1.18 (0.98–1.42) (150 bid)	4.9 vs. 4.4 1.11 (0.92–1.34)	3.3 vs. 5.2 0.64 (0.52–0.79)	4.0 vs. 4.8 0.83 (0.70–0.89)
Gastrointestinal bleedings in patients ≥75 years				
Event rates (DOACs vs. VKAs %/years) HR (or RR for dabigatran) (IC 95%)	2.2 (110 mg bid) vs. 1.6 2.8 (150 mg bid) vs. 1.6 1.39 (1.03–1.98) (110 bid) 1.79 (1.35–2.37) (150 bid)	2.8 vs. 1.7 1.69 (1.19–2.39)	1.3 vs. 1.3 0.99 (0.69–1.42)	2.2 vs. 1.7 1.32 (1.01–1.72)
Intracranial bleeding in patients ≥75 years				
Event rates (DOACs vs. VKAs %/years) HR (or RR for dabigatran) (IC 95%)	0.37 (110 mg bid) vs. 1 0.41 (150 mg bid) vs. 1 0.37 (0.21–0.64) (110 bid) 0.42 (0.25–0.70) (150 bid)	0.66 vs. 0.83 0.80 (0.50–1.28)	0.43 vs. 1.29 0.34 (0.20–0.57)	0.5 vs. 1.2 0.40 (0.26–0.62)

## 3. DOACs and Frailty

“Frailty” is defined as a vulnerability to infectious processes and physical and emotional stresses [27]. The prevalence of such a condition increases with age and ranges from 9% in 75–79-year-old patients to 26% in patients ≥85 years [28].

Frail subjects are less likely to receive OAC despite evidence supporting the use of OAC in this population [15,29]. According to the results of RCTs [30,31], meta-analyses [4,21], and large registries [15,24,32], when compared to warfarin, DOACs demonstrate a better risk-benefit profile in frail patients [11,24,30,33,34]. The prescription of a reduced dose of OAC is less effective in preventing adverse outcomes [35,36,37].

In the systematic review by Oqab et al. [38], it was highlighted that 40% of hospitalized elderly patients with AF (over the age of 80) were classified as frail, and the rate of OAC prescription was lower in these patients than in non-frail patients (OR 0.49, 95% CI 0.32–0.74) [38]. Among frailty characteristics, cognitive impairment, malnutrition risk, depression, and falls were recognized as the main reasons for the under-prescription of oral anticoagulants [38].

The FRAIL-AF study [39] showed that severely frail patients are much less likely to be prescribed with DOACs than non-frail, mildly or moderately frail patients (OR 3.4), regardless of the individual patient’s thrombotic and bleeding risk. This evidence suggests that frailty, in clinical practice, significantly influences the prescription of DOACs [39].

Thus, the best antithrombotic therapeutic strategy in frail AF patients remains unclear at present. Moreover, comparison studies between different DOACs are still not available. Anyhow, apixaban seems to have a good risk/benefit profile in older patients, especially in those with renal failure [40,41,42]. On the other hand, it could be reasonable to avoid dabigatran and rivaroxaban because of the increased risk of gastrointestinal bleeding described in patients aged ≥75 years [43].

## 4. DOACs in Patients with AF and Active Cancer

AF is commonly diagnosed in the setting of active cancer [44]. Antithrombotic prevention against the risk of cerebral stroke and systemic embolism in patients with AF and cancer disease is essential [45], and the risk of bleeding also depends on the type of tumor [46].

Due to the low life expectancy and high bleeding risk of cancer patients, the major RCTs of DOACs have included few patients with AF and cancer [12,17,47,48]. Therefore, data on this sub-population remain lacking and uncertain. In this regard, several observational studies and meta-analyses have investigated the efficacy and safety of DOACs in this population [49,50,51,52], assessing their viability when compared to warfarin [49,51,52] (Table 2).

Russo et al. published a systematic review of the literature of six eligible studies, founding that the efficacy and safety of DOACs in cancer patients are similar to that of the general population [49]. In particular, authors found a low annual incidence of bleeding and thrombotic events in cancer patients treated with DOACs compared to those treated with warfarin. Moreover, the risk of such events was comparable to non-cancer patients regardless of the treatment used (DOACs or VKAs) [49].

A systematic review and metanalysis of three sub-studies of ARISTOTLE [53], ROCKET-AF [54], and ENGAGE-TIMI 48 trials [55] showed no significant differences in safety and efficacy outcomes between cancer and non-cancer patients in OAC therapy (all *p* < 0.05) [51]. Moreover, DOAC therapy resulted in a significantly lower risk of stroke/systemic embolism (*p* = 0.04), venous thromboembolism (*p* < 0.0001), and a decreased risk of intracranial or gastrointestinal bleeding compared with warfarin (*p* = 0.04) [51].

Yang et al., in a network meta-analysis [52], showed that in AF patients with cancer, apixaban was associated with the lowest risk of stroke/systemic embolism (OR 0.12, 95% confidence interval [CI] 0.05–0.52), followed by rivaroxaban, dabigatran, edoxaban, and warfarin. Apixaban was also the best treatment option to avoid major bleeding, followed by dabigatran and edoxaban (OR 0.39, 95% CI 0.18–0.79) [52].

A large meta-analysis (46,424 DOAC users and 182,797 VKA users) comparing the efficacy and safety of DOACs and VKAs in cancer patients has shown that DOACs are more effective in preventing strokes in the course of the AF [56]. Indeed, DOACs, compared to VKAs, significantly reduced the risk of both ischemic (RR 0.84; *p* = 0.007) and hemorrhagic stroke (RR 0.61, *p* < 0.00001) [56]. Moreover, the risk of major bleeding was significantly reduced by up to 32% (RR 0.68, *p* = 0.01) and the risk of systemic embolism or any type of stroke by 35% with DOACs compared with VKA (RR 0.65, *p* < 0.0001) [56]. Similarly, the use of DOACs versus VKAs significantly reduced the risk of intracranial or gastrointestinal bleeding (RR 0.64, *p* = 0.006) [56].

Furthermore, in a study of 16,096 patients with AF and active cancer [57], the bleeding rate was similar with rivaroxaban and dabigatran but significantly lower with apixaban (HR 0.37; *p* = 0.002). None of the anticoagulants showed greater efficacy in reducing the incidence of ischemic stroke [57].

The Scientific and Standardization Committee (SSC) of the International Society on Thrombosis and Haemostasis (ISTH) recommends that individual decisions should be made for a patient with cancer and AF, considering the risk of stroke and bleeding [58]. In patients who had initiated anticoagulation before anti-cancer treatment, therapy should not be modified if there are no significant interactions with oncological drugs. In the case of newly diagnosed AF during chemotherapy, DOACs should be preferred over VKAs or low-molecular-weight heparin (LMWH) if no significant drug–drug interactions are found. The exception is patients with gastrointestinal neoplasms or other gastrointestinal tract diseases predisposing to bleeding, where the OAC prescription should be strongly individualized. LMWH in therapeutic doses should only be recommended when the patient cannot take oral anticoagulants. VKAs are recommended in patients with mechanical heart valves or moderate-to-severe mitral stenosis.

In conclusion, several preliminary pieces of evidence suggest that DOACs are effective and safe in cancer patients with AF, but RCTs should improve these findings. The choice of the DOAC should be individualized, considering the prothrombotic risk related to cancer disease and the risk of bleeding.

**Table 2 biomedicines-11-00131-t002:** Results of the main studies exploring the efficacy and safety of direct oral anticoagulants in cancer patients with atrial fibrillation.

Authors, Reference	Main Study Characteristics	Ischemic Events	Major Bleeding	Conclusions
Russo et al. [49]	Systematic Review of retrospective studies (6 studies included) DOACs in AF cancer patients Cancer Vs. Non-cancer patients	Annual incidence range 0 to 4.9% versus 1.3 to 5.1%	Annual incidence range 1.2 to 4.4% versus 1.215 to 3.1%	No significant differences in safety and efficacy outcomes between cancer and no-cancer patients with AF on DOACs
Deng et al. [51]	Systematic Review and Meta-Analysis (5 studies included) DOACs Vs. Warfarin	SSE	Intracranial or GI	In cancer patients, DOACs have similar rates or lower rates of ischemic and bleeding events and a reduced risk of venous thromboembolism compared with warfarin.
		RR = 0.52 95% CI, 0.28–0.99 *p* = 0.04	RR = 0.65 95% CI, 0.42–0.98 *p* = 0.04	
		VTE	MB	
		RR = 0.37 95% CI, 0.22–0.63 *p* < 0.0001	RR = 0.73 95% CI, 0.53–1.00 *p* = 0.05	
		IS	MB or CRNMB	
		RR = 0.63 95% CI, 0.40–1.00 *p* = 0.05	RR = 1.00 95% CI, 0.86–1.17 *p* = 0.96	
		MI	Any bleeding	
		RR = 0.75 95% CI, 0.45–1.25 *p* = 0.26	RR = 0.93; 95% CI, 0.78–1.10 *p* = 0.39	
Mariani et al. [56]	Meta-analysis (9 studies included) DOACs versus Warfarin	SSE	HS	In patients with cancer and non-valvular AF, the use of DOACs is associated with a significant reduction of thromboembolic and bleeding events in patients when compared to warfarin
		RR 0.65 95% CI, 0.52–0.81 *p* = 0.001	RR 0.61 95% CI 0.52–0.71 *p* = 0.00001	
		IS	MB	
		RR 0.84 95% CI 0.74–0.95 *p* = 0.007	RR 0.68 95% CI 0.50–0.92 *p* = 0.01	
		MI	Intracranial or GI	
		RR 0.71 95% CI 0.48–1.04 *p* = 0.08	RR 0.64 95% CI 0.47–0.88 *p* = 0.006	
			MB or CRNMB	
			RR 0.94; 95% CI 0.78–1.13; p 0.50	
			Any bleeding	
			RR 0.91 95% CI 0.78–1.06 *p* = 0.24	

DOACs: direct oral anticoagulants; AF: atrial fibrillation; SSE: stroke/systemic embolism; VTE: venous thromboembolism; IS: ischemic stroke; MI: myocardial infarction; GI: gastrointestinal; MB: major bleeding; CRNMB: clinically relevant non-major bleeding; HS: hemorrhagic stroke.

## 5. DOAC Treatment in Obese Patients

In both ENGAGE AF-TIMI [48] and ARISTOTLE [17] trials, an important amount of the enrolled subjects were overweight, and approximately 40% of patients were obese (BMI ≥ 30 kg/m^2^).

According to the European Society of Cardiology guidelines for the management of the AF [1], obesity is a comorbidity that needs to be corrected as part of the ABC-integrated approach. At the extreme of the distribution of obese patients, there are underweight patients, whose prevalence is higher in Asia [59,60,61], with a limited representation in multicenter trials that validated DOACs [62,63].

Studies that have investigated the efficacy and safety of DOACs in obese patients are summarized in Table 3.

The post-hoc analysis of the ROCKET-AF trial evaluated patients with normal BMI (BMI from 18.5 to <25 kg/m^2^), overweight (BMI 25 to <30 kg/m^2^), or obese (BMI ≥ 30 kg/m^2^), and overall, 36.5% of subjects enrolled were classifiable as obese [64]. The risk of stroke was statistically significantly lower for obese patients with BMI ≥ 35 kg/m^2^ than that for normal-weight patients in both the rivaroxaban and warfarin group [63].

The post-hoc analysis of the ARISTOTLE trial found that 39.4% of enrolled patients had a BMI ≥ 30 kg/m^2^. In these patients, the comparison between apixaban and warfarin showed no differences in terms of stroke or major bleeding [61].

In the ENGAGE AF-TIMI 48 trial, a higher BMI value was associated with a lower risk of stroke or thromboembolism and better survival but with an increased risk of bleeding [62]. The efficacy and safety profile of edoxaban was comparable among BMI categories ranging from 18.5 to >40 kg/m^2^, indicating the reliability of edoxaban treatment also in patients with obesity [62].

A systematic review and meta-analysis based on trials about various levels of BMI showed that in patients with soft or morbid obesity (class III, with BMI 40–49 kg/m^2^), there are limited data on the efficacy and safety of dabigatran and rivaroxaban, while more data are available for edoxaban and apixaban [64].

In extremely obese (BMI ≥ 50 kg/m^2^), data are very limited for all DOACs [65]. The assessment of DOAC plasma levels, as well as the evaluation of the effect on coagulation parameters, could improve the management in obese patients [66,67,68,69,70]. According to the International Society of Thrombosis and Haemostasis, VKAs should be the treatment of choice in morbidly obese patients [71]. Both the International Society of Thrombosis and Haemostasis and the European Heart Rhythm Association suggest that if a DOAC is administered in morbidly obese patients with AF, drug-specific peak and trough levels (anti-FXa for apixaban, edoxaban, and rivaroxaban, ecarin time or dilute-thrombin time with appropriate calibrators for dabigatran, or mass spectrometry drug level for any of the DOACs) should be checked to switch to VKAs if any drug level is found below the expected range [71,72].

In the ENGAGE AF-TIMI 48 trial, the concentration of edoxaban and the anti-factor Xa activity at baseline after one month of treatment was assessed in a large number of patients [4]. A sub-analysis of ENGAGE AF-TIMI 48 study showed that the concentrations of edoxaban and anti-factor Xa activity did not vary significantly between underweight, normal BMI, and obesity categories [62]. The data of the ENGAGE AF-TIMI 48 trials were also analyzed in ratio to weight in a study that evaluated patients with weight ≤ 55 kg and ≥120 kg in comparison with patients weighing 80–84 kg, highlighting that the concentrations of edoxaban, the activity anti-factor Xa, and factor Xa inhibition rates were comparable between the three groups [73].

The peak concentrations of DOACs (apixaban, dabigatran, and rivaroxaban) were measured in a cohort of 38 obese patients (median weight, 135.5 kg) with venous thromboembolism or AF by Piran et al. [74]. Results showed that the peak drug concentration was higher than the expected median trough level in the majority of patients (79%); however, a considerable part of the study population (21%) still had a peak plasma concentration below the usual on-therapy range of peak concentration for the corresponding DOACs [74].

In a cohort of 100 patients with AF or venous thromboembolism and weight > 120 kg receiving apixaban or rivaroxaban, Martin et al. found no significant relationship between DOAC concentrations at peak or trough, weight, BMI, or renal function [72].

In the study of Russo et al. [67], among 58 patients with extreme obesity (BMI ≥ 40 kg/m^2^) and AF, nine patients (15.5%) showed that DOAC plasma concentrations were out of the expected ranges. Among these patients, according to the multivariate logistic analysis, the only independent predictor of DOAC plasma levels out of the expected ranges was the inappropriate prescription of low-dose DOACs (hazard ratio = 29.37; *p* = 0.0002) [67]. According to these results, extreme obesity does not significantly impact DOAC plasma levels [67].

## 6. DOAC Use in Patients with Low Body Weight

In the RE-LY trial, the study population was stratified according to groups of body weight to analyze the safety and efficacy of dabigatran in each subgroup of patients [12]. Both dabigatran 150 mg BID and 110 mg BID showed greater efficacy and similar safety compared to warfarin in patients < 50 kg (*n* = 376) [12].

Even if low body weight increases the exposure to rivaroxaban, in a single-blind, placebo-controlled study, no differences in plasma levels, nor the incidence of adverse events related to the body weight (range 45–173 kg), were found in healthy volunteers of both genders with the use of a fixed dose of rivaroxaban (10 mg, half the dose recommended for stroke prevention in AF) [75].

In the ARISTOTLE trial, despite a higher risk of stroke or systemic embolism and major bleeding, patients with isolated body weight < 60 kg (or age > 80, or serum creatinine > 1.5 mg/dL) showed better outcomes with apixaban 5 mg BID versus warfarin when compared with patients without these characteristics [17].

In both phase III studies, edoxaban half standard dose (30 mg) was administered to participants weighing ≤ 60 kg [48,76] because of the risk of an increased drug plasma level in these patients [48].

Underweight patients (BMI < 18.5 kg/m^2^) suffer from a higher risk of bleeding and all-cause death, as demonstrated in a recent survey from Korea (underweight patients vs. normal weight patients: adjusted HR 4.135 *p* = 0.008; adjusted HR 10.524, *p* < 0.001) [77]. However, there was no significant difference in the risk of thromboembolism among these groups [77].

Russo et al. [78] conducted a propensity score-matched study on elderly AF patients (>75 years) with a low body weight (<60 kg) sourced from the Italian cohorts of PREFER in AF and PREFER in AF PROLONGATION registries to compare the safety and effectiveness of DOACs versus VKA therapy (213 patients in each group). No statistically significant differences were found in any analyzed outcome (DOACs vs. VKAs; thromboembolic events: 3.76% vs. 4.69%, *p* = 0.63; major bleeding events: 1.88% vs. 4.22%, *p* = 0.15; hospitalizations: 9.9% vs. 16.9%, *p* = 0.06) [78]. However, a net clinical benefit (+1.6) of DOACs vs. VKAs was demonstrated [78].

Furthermore, an analysis of 279 AF patients aged ≥ 80 years and weighed ≤ 60 kg (136 in DOAC vs. 143 in VKA group) showed a lower incidence of all-cause of mortality in the DOAC group (14.91 per 100 person-years in DOAC vs. 37.94 per 100 person-years in VKA group, adjusted HR 0.43; *p* = 0.003) with no significant differences in major bleeding events (9 in DOAC vs. 13 in VKA group, *p* = 0.6), suggesting the safety and efficacy of the use of DOACs in octogenarians with low weight [79].

In conclusion, patients with low body weight (<60 kg) were poorly included in RCTs. In each RCT, subgroup analyses showed that the efficacy and safety of DOACs demonstrated in patients weighing >60 kg is maintained in patients with low body weight, with edoxaban requiring a dose reduction, as well as apixaban, if at least one other clinical criterion is met. Anyway, monitoring drug levels is a prudent strategy in very low body weight patients.

## 7. DOACs in Patients with AF and BHV

To date, it is still unclear which is the best treatment option for AF patients with BHVs [80]. Current guidelines recommend lifelong OAC therapy in this subgroup of patients with a class IC of evidence [81], with VKAs preferred to DOACs for the first three months after the procedure [81,82]. ESC/EACTS guidelines [82] admit the use of rivaroxaban directly after surgical BHV implantation in the mitral position (class IIb C) due to the results of the Rivaroxaban for Valvular heart disease and atRial fibrillation (RIVER) trial [83].

Some experiences with DOACs are available in clinical practice in this setting.

Yadlapti et al. registered almost no thrombotic events (one transient ischemic attack) but a higher risk of bleeding (five major bleedings, six minor bleedings, and one hemorrhagic stroke) in a cohort of 73 AF patients treated with DOACs (dabigatran, *n* = 44; rivaroxaban, *n* = 25; apixaban, *n* = 4) after aortic (*n* = 61) or mitral (*n* = 12) BHV replacement [84]. However, 72% of patients were on antiplatelet treatment, possibly causing a bias in the incidence of bleeding outcomes [85].

Indeed, in a larger, multicenter observational study, Russo et al. [84] recorded two thromboembolic events and only four major bleedings in a cohort of 122 AF patients with a prior BHV replacement or valve repair, treated with apixaban (53.1%), dabigatran (31%), or rivaroxaban (15.5%), with only 20% treated with antiplatelet therapy also [86]. Table 4 summarizes the results of these studies.

Of the four major RCTs comparing DOACs to warfarin [12,17,47,77], only the ENGAGE AF-TIMI 48 trial and the ARISTOTLE trial included AF patients with BHV replacement. In a post-hoc analysis from the ENGAGE AF-TIMI 48, the analysis of 191 AF patients with aortic (31.4%) or mitral (68.6%) BHV replacement, edoxaban showed a comparable rate of stroke/systemic embolism (HR, 0.37; *p* = 0.15) and major bleeding (HR, 0.5; *p* = 0.26) [86] but lower rates of myocardial infarction, stroke, or cardiovascular death (HR, 0.36; *p* = 0.03) when compared to warfarin [86].

Moreover, in the ARISTOTLE trial, no significant differences were found in safety or efficacy outcomes between apixaban and warfarin in the included cohort of 156 patients with AF and BHVs or valve repair [83].

These results were consistent with those of two multicenter observational studies [87,88], in which DOACs also showed a lower rate of major bleeding compared to warfarin. Table 5 displays the results of the aforementioned observational studies.

Regarding RCTs, the DAWA (Dabigatran Versus Warfarin After Mitral and/or Aortic Bioprosthesis Replacement and Atrial Fibrillation Postoperatively) pilot study was the first RCT designed to compare efficacy and safety outcomes of dabigatran 110 mg BID to warfarin in patients with mitral and aortic BHV replacement [89]. In a follow-up period of 90 days, no differences in the measured outcomes were recorded between the two treatments, but the trial was prematurely terminated because of the low enrollment (34 patients) [89].

In a recent small trial on 50 AF patients undergoing aortic BHV replacement, apixaban (*n* = 25) and warfarin (*n* = 25) were compared for adverse events in three months following surgery. Only one CV death due to massive pericardial effusion was recorded nine days after surgery in the warfarin group. At the end of the follow-up period, three patients experienced a major bleeding event in the warfarin group, with no such events in the apixaban one. In light of these pieces of evidence, authors concluded that apixaban is proven safer than warfarin early after aortic BHV replacement with comparable efficacy in preventing valvular dysfunction (no event recorded in either group) [90].

Recently, 1005 AF patients were enrolled in The RIVER trial [91] to randomly receive rivaroxaban or warfarin after surgical mitral BHV replacement. No differences in ischemic and bleeding events or death occurred between rivaroxaban versus warfarin at the end of the follow-up (12 months follow-up; stroke, 3% vs. 2.4; major bleeding, 1.4% vs. 2.6%; death, 4% vs. 4%). Table 6 summarizes the results of these latter RCTs.

Data on DOAC clinical profiles in patients with transcatheter aortic valve implantation (TAVI) and AF are still lacking. According to the results of the propensity score-matched study of Jochheim et al. [91], DOAC safety was comparable to warfarin at the cost of a higher incidence of the composite efficacy endpoint (all-cause mortality, myocardial infarction, and any cerebrovascular events; 21.2% vs. 15%; HR, 1.44; *p* = 0.05) in a cohort of 962 TAVI patients with AF discharged on DOACs (*n* = 326; 53.7% rivaroxaban, 39.2% apixaban, and 7.1% dabigatran) or warfarin (*n* = 626), and followed-up for 12 months [91] (Table 7).

The prospective multicenter observational Optimized Transcatheter Valvular Intervention (OCEAN) study [93] showed a low incidence of all-cause mortality in DOAC patients (*n* = 227) compared to VKAs (*n* = 176) in TAVI patients with AF (10.3% vs. 23.3%; HR, 0.391; *p* = 0.005) during a median follow-up of 568 days [93] (Table 7). The same conclusions were applied in the three-year follow-up study of Butt et al. [94] for the incidence of arterial thromboembolism, bleeding, or mortality. The study included 219 (29.8%) AF patients treated with DOACs and 516 (70.2%) treated with VKAs following TAVI (Table 7).

Regarding RCTs, in the ENVISAGE trial [95], when compared to VKAs, edoxaban was non-inferior for the composite endpoint of death from any cause, myocardial infarction, ischemic stroke, systemic thromboembolism, and valve thrombosis (HR, 1.05; *p* = 0.01), as well as for major bleeding (HR, 1.05; *p* = 0.01), but the rate of major bleeding events was found higher in the DOAC group (HR, 1.40; *p* = 0.93), because of a higher rate of gastrointestinal bleeding [95] (Table 7).

Also, in the AF cohort of the ATLANTIS trial (stratum 1) (Anti-Thrombotic Strategy to Lower All cardiovascular and Neurologic Ischemic and Hemorrhagic Events after Trans-Aortic Valve Implantation for Aortic Stenosis), apixaban 5 mg BID compared to warfarin was not superior for both the primary and safety outcomes (primary efficacy outcome: HR, 1.02; 95% CI, 0.68 to 1.51; primary safety outcome: HR, 0.91; 95% CI, 0.52 to 1.60) [97] (Table 7).

In conclusion, guidelines recommend OAC alone therapy in AF patients undergoing BHV replacement [82]. Several studies highlight the favorable role of DOACs over VKAs among AF patients undergoing surgical BHV replacement. However, those studied are limited by a large use of concomitant antiplatelet therapy, which implies biases both for thromboembolic and bleeding outcomes. Except for mitral valve replacement, about the early use of DOACs after BHV replacement (first three months), data are scarce to draw definitive conclusions. Among AF patients undergoing TAVI, OAC alone is preferred over OAC plus clopidogrel [81,94,96,98], but conclusions on the DOAC profile are unclear at this point.

## 8. Conclusions

OAC therapy in patients with AF should be based on the risk of thromboembolism, stroke, and bleeding but also on the patient’s preference. Special populations require careful evaluation and personalized therapy, considering the evidence regarding the pathology, comorbidities, and the risk–benefit ratio of long-term OAC. Many studies have been performed to test the efficacy and safety of DOACs, but special populations—elderly, frail, patients with extreme weight, cancer patients, and subjects with BHV—are underrepresented in the pivotal RCTs. “Real-world-setting” studies help to shed light on the OAC management of these patients; however, there is a need for further studies in this area.

## Figures and Tables

**Table 3 biomedicines-11-00131-t003:** Main studies that investigated the efficacy and safety of DOACs in obese patients.

First Author	Study Design	Patients (*n*)	BMI (kg/m^2^)	Stroke/SE HR (CI 95%)	Major Bleeding HR (CI 95%)	Follow-Up (Mean in Years)
Sandhu et al. [61]	RCT Apixaban * vs. Warfarin (ARISTOTLE sub-analysis)	17,913	18–<25 (*n* = 4052) 25–30 (*n* = 6702) ≥30 (*n* = 7159)	0.86 (0.68–1.08) 0.79 (0.61–1.02)	0.82 (0.68–0.99) 0.91 (0.74–1.10)	1.8 **
Boriani et al. [62]	RCT Edoxaban * vs. Warfarin (ENGAGE-TIMI 48 sub-analysis)	21,105	18–25 (*n* = 4491) 25–<30 (*n* = 7903) 30–<35 (*n* = 5209) 35–40 (*n* = 2099) ≥40 (*n* = 1149)	0.91 (0.78–1.07) 0.82 (0.68–1.00) 0.68 (0.52–0.89) 0.54 (0.35–0.83)	1.03 (0.88–1.20) 1.12 (0.94–1.34) 1.18 (0.94–1.48) 1.28 (0.96–1.70)	2.8 **
Balla et al. [63]	RCT Rivaroxaban * vs. Warfarin (ROCKET-AF post-hoc analysis)	14,264	18.5–24.99 (*n* = 3289) 24–29.99 (*n* = 5535) ≥30 (*n* = 5206)	0.78 (0.64–0.96) 0.65 (0.52–0.80)	0.99 (0.82–1.18) 0.91 (0.75–1.10)	2
Kido et al. [64]	Meta-analysis of 6 studies DOACs vs. Warfarin.	8732	>40	0.85 (0.60–1.19)°	0.63 (0.43–0.94)°	-

DOACs = direct oral anticoagulants; ClCr = clearance of creatinine; HR = Hazard ratio; CI = confidence interval; SE = systemic embolism; BMI = body mass index. ****** median; °ODDS RATIO. * reduced dose for apixaban: age ≥ 80 years, body weight ≤ 60 kg, or serum creatinine level ≥ 1.5 mg/dL; reduced dose for edoxaban: CrCl ≤ 50 mL/min, a body weight ≤ 60 kg use of P-glycoprotein inhibitors; reduced dose for rivaroxaban: ClCr 30–49 mL/min.

**Table 4 biomedicines-11-00131-t004:** Observational studies on the clinical performance of Direct oral anticoagulants after bioprosthetic heart valve replacement in atrial fibrillation patients.

First Author, Reference	Study Characteristics (Design, Included patients, Procedure Included)	DOAC n, (%)	Ischemic Events (%)	Bleeding Events (%)
Yadlapati et al. [85]	Single center Retrospective Observational 73 patients included ABHV; MBHV	Dabigatran 44, (60.3) Rivaroxaban 25, (34.2) Apixaban 4, (5.5)	1 TIA (1.4)	5 MB (6.9) 6 mB (8.2), 2 ICH (2.7)
Russo et al. [84]	Multicenter Retrospective Observational 122 patients included ABHV; MBHV; VR	Dabigatran 38, (31) Rivaroxaban 19, (15.5) Apixaban 65, (53.3)	TE 2 (1,7) M.A.I: 0.8%	4 (3.3) M.A.I: 1.3%

DOACs: direct oral anticoagulant; ABHV: aortic bioprosthetic heart valve; MBHV: mitral bioprosthetic heart valve; TIA: transient ischemic attack; MB: major bleeding: mB: minor bleeding; VR: valve repair; TE: thromboembolic events; M.A.I: mean annual incidence.

**Table 5 biomedicines-11-00131-t005:** Overview of the study characteristics comparing Direct oral anticoagulants with Vitamin K antagonist oral anticoagulants in AF patients with bioprosthetic valves or prior surgical valve repair.

Author, Reference	Study Characteristics (Design, Number of Patients, Procedure Included)	DOAC n, (%)	Results
Carnicelli et al. [86]	Post-hoc analysis phase III trial 191 patients included ABHV; MBHV	Edoxaban 121 (63.4)	Efficacy outcome
			S/SE HDE vs. warfarin: HR 0.37 95% CI, 0.10–1.42 *p* = 0.15 LDE vs. warfarin: HR 0.53 95% CI, 0.16–1.78 *p* = 0.31
			Safety outcome
			MB HDE vs. warfarin: HR 0.5, 95% CI 0.15–1.67 *p* = 0.26 LDE vs. warfarin HR 0.12, 95% CI 0.01–0.95 *p* = 0.045
			Primary net clinical outcome (S/SE, MB, death)
			HDE vs. warfarin: HR 0.46 (95% CI, 0.23–0.91) *p* = 0.03 LDE vs. warfarin HR (0.43, 95% CI, 0.21–0.88) *p* = 0.02
Guimarães et al. [83]	Post-hoc analysis phase III trial 156 patients included ABHV; MBHV; VR	Apixaban 87 (55.8)	Efficacy outcomes
			S/SE HR 1.714 (95% CI 0.313–9.372) *p* = 0.53
			ACS HR 1.714 (95% CI 0.313–9.372) *p* = 0.53
			IS HR 3.286 (95% CI 0.37–29.4) *p* = 0.29
			MI HR 0.825 (95% CI 0.367–29.40) *p* = 0.29
			Safety outcome
			MB HR 0.882 (95% CI 0.309–2.519) *p* = 0.82
			MB/CRNMB HR 0.781 (95% CI 0.317–1.925) *p* = 0.59
			ICH HR 0.467 (95% CI 0.042–5.187) *p* = 0.54
			GI bleeding HR 1.244 (95% CI 0.208–7.448) *p* = 0.81
			Any bleeding HR 0.866 (95% CI 0.517–1.451) *p* = 0.59
Russo et al. [85]	Retrospective Propensity Score matched 130 for each group ABHV; MBHV	Apixaban 72 (55.4) Rivaroxaban 39 (30.0) Dabigatran 17 (13.1) Edoxaban 2 (1.4)	Efficacy outcome
			S/SE-TIA HR 0.49 (95% CI, 0.19–1.22) *p* = 0.14
			Safety outcome
			MB HR 0.59 (95% CI, 0.15–2.4) *p* = 0.47
			ICH HR 0.33 (95% CI, 0.05–2.34) *p* = 0.3
Duan et al. [87]	Retrospective cohort study 2672 patients included ABHV; MBHV	Dabigatran 362 (13.5) Apixaban 60 (2.2) Rivaroxaban 17 (0.6)	Efficacy outcome
			Composite of IS/TIA/SE HR 1.19 (95% CI, 0.96–1.48), *p* = 0.106
			Safety outcome
			Composite of MB ^1^ HR 0.69 (95% CI 0.56–0.85) *p* < 0.001

DOACs: direct oral anticoagulant; ABHV: aortic bioprosthetic heart valve; MBHV: mitral bioprosthetic heart valve; S/SE: stroke/systemic embolism; HDE: high-dose edoxaban; LDE: low-dose edoxaban; VR: valve repair; HR: hazard ratio; CI: confidential interval; MB: major bleeding; CRNMB: clinically relevant non-major bleeding; ACS all-cause stroke; IS: ischemic stroke; MI: myocardial infarction; ICH: intracranial hemorrhage; GI: gastrointestinal; TIA: transient ischemic attack; ^1^: Gastrointestinal bleeding, intracranial hemorrhage and bleeding from other sites.

**Table 6 biomedicines-11-00131-t006:** Characteristics of the randomized clinical trials comparing Direct oral anticoagulants with vitamin K antagonist oral anticoagulants in AF patients with bioprosthetic valves or surgical valve repair.

Author, Reference	Study Design	Number of Patients (DOACs/VKAs)	Results
Durães et al. [92]	Phase 2 RCT pilot study Dabigatran 110 mg Vs. Warfarin ABHV; MBHV	27 patients included Dabigatran = 15 Warfarin = 12	New intracardiac thrombus Warfarin = 1 Dabigatran = 0 RR 1.1, CI 95% 0.9–1.3 *p* = 0.42
Piepiorka-Broniecka et al. [89]	Prospective RCT Apixaban Vs. Warfarin ABHV	50 patients included Apixaban = 25 Warfarin = 25	Cumulative Death Warfarin = 1 Apixaban = 0 *p* = 0.31 Cumulative Bleeding Warfarin = 3 Apixaban = 0 *p* = 0.07 Valve dysfunction Warfarin = 0 Apixaban = 0
Guimarães et al. [90]	Multicenter RCT Rivaroxaban Vs. Warfarin MBHV	1005 patients included Rivaroxaban = 500 Warfarin = 505	Efficacy outcome
			^1^ Composite of Death/MACE ^2/^MB ^3^ Warfarin = 340.1 Rivaroxaban = 347.5 RMST difference: 7.4 days (−1.4–16.3) *p* < 0.001 for noninferiority *p* = 0.10 for superiority
			Safety outcome
			MB Warfarin = 13 Rivaroxaban = 7 HR 0.54 (0.21–1.35) *p* = N/A
			ICH Warfarin = 5 Rivaroxaban = 0 N/A
			Fatal bleeding Warfarin = 2 Rivaroxaban = 0 N/A

DOACs: direct oral anticoagulants; VKAs: vitamin K antagonists oral anticoagulant; ABHV: aortic bioprosthetic heart valve; MBHV: mitral bioprosthetic heart valve; RCT: randomized clinical trial; RR: relative risk; MB: major bleeding; ICH: intracranial hemorrhage; RMST: restricted mean survival time; ^1^: mean time until a primary-outcome event in days; ^2^: ischemic attack, valve thrombosis, systemic embolism not related to the central nervous system, or hospitalization for heart failure; ^3^: according to the criteria of the Rivaroxaban Once Daily Oral Direct Factor Xa Inhibition Compared with Vitamin K Antagonism for Prevention of Stroke and Embolism Trial in Atrial Fibrillation (ROCKET AF): Any bleeding; Major bleeding; Intracranial bleeding; Fatal bleeding; Clinically relevant nonmajor bleeding; Minor bleeding; N/A: not available.

**Table 7 biomedicines-11-00131-t007:** Characteristics and results of the studies comparing Direct oral anticoagulants with vitamin K antagonist oral anticoagulants in AF patients after TAVI.

Author, Reference	Study Design	Number of Patients (NOACs/VKAs)	Results
Jochheim et al. [91]	Prospective Observational Multicenter study DOACs Vs. VKAs	962 patients included DOACs = 326 VKAs = 636	All-cause mortality/MI/CVE DOACs = 63/326 VKAs = 87/636 1.44, CI 95% 1.00–2.07 *p* = 0.050
			All-cause death DOACs = 47/326 VKAs = 70/636 1.36, CI 95% 0.90–2.06 *p* = 0.136
			MB DOACs = 67/326 VKAs = 144/636 0.9, CI 95% 0.64–1.26 *p* = 0.548
			MACCE DOACs = 30/326 VKAs = 42/636 1.43, CI 95% 0.85–2.43 *p* = 0.173
Kawashima et al. [93]	Prospective observational Multicenter DOACs Vs. VKAs	403 patients included DOACs = 227 VKAs = 127	All-cause mortality 10.3% vs. 23.3%; HR: 0.391, CI 95% 0.204–0.749; *p* = 0.005
Butt et al. [94]	Retrospective Observational Cohort study DOACs Vs. VKAs	735 patients included DOACs = 219 VKAs = 516	Arterial thromboembolism ^1^ HR, 1.23; 95% CI, 0.58–2.59
			Bleeding HR, 1.14; *p* = 0.63–2.06
			All-cause mortality HR, 0.93, 95% CI, 0.61–1.4
Van Mieghem et al. [95]	RCT Edoxaban Vs. VKAs	1426 patients included Edoxaban = 713 VKAs = 713	Composite primary efficacy outcome ^2^ HR, 1.05, 95% CI, 0.85 to 1.31; *p* = 0.01 for noninferiority
			MB HR, 1.40; 95% CI, 1.03 to 1.91 *p* = 0.93 for noninferiority
Collet et al. [96]	RCT (stratum 1—with indication for OAC) Apixaban Vs. VKAs	451 included patients In stratum 1 Apixaban = 223 VKAs = 228	Primary efficacy outcome ^3^ HR, 1.02; 95% CI, 0.68 to 1.51
			Primary safety outcome ^4^ HR, 0.91; 95% CI, 0.52 to 1.60

DOACs: direct oral anticoagulants; VKAs: vitamin K antagonist oral anticoagulant; MI: myocardial infarction; CVE: cerebrovascular events; MB: major bleeding; MACCE: major adverse cardiac and cerebrovascular events; RCT: randomized clinical trial; OAC: oral anticoagulant; HR: hazard ratio; CI: confidential interval; ^1^ composite of ischemic stroke, transient cerebral ischemia, and thrombosis or embolism in peripheral arteries); ^2^ death from any cause, myocardial infarction, ischemic stroke, systemic thromboembolic event, valve thrombosis, or major bleeding; ^3^ composite of death, myocardial infarction, stroke or transient ischemic attack, systemic embolism, intracardiac or bioprosthetic thrombosis, deep vein thrombosis or pulmonary embolism, and life-threatening, disabling, or major bleeding over 1-year follow-up; ^4^ major disabling, or life-threatening bleeding.

## Data Availability

Not applicable.

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
