# Peer review of "Direct Oral Anticoagulants for Stroke Prevention in Special Populations: Beyond the Clinical Trials"

_biomedicines, 2023, doi:10.3390/biomedicines11010131_

Round 1
Reviewer 1 Report
DOACs are now the recommended preventative therapy for NVAF and guidelines recommend their use. The authors of this review argue that the trials on which the evidence for the guidelines is based exclude patients in special populations including older, ageing, frail and cancer patients with AF. While this is a worthy topic the narrative review presented does not present a clear picture of the role of DOACs in these populations. The review treats each population separately however the AF population are frequently old and frail and with high BMI hence it is difficult to separate out the contribution of each individual factor.
The standard of English is very poor with frequent language and grammatical errors which make the article difficult to follow. There is an absence of any summary tables of the studies discussed and there is no interpretation of the data and its implications for practice apart from the very brief and general conclusion at the end.
I have the following specific comments:
1. Title: The title should reflect the topic and should mention patients with AF. The current title suggests that the study includes all types of patients using DOACs.
2. The introductory paragraph should include a description of the current guidelines to set the subsequent sections in context.
3. For each special population a summary table of the trials discussed would be useful focussing on the main outcome variables of stroke and bleeding
4. Many aged patients are also frail, therefore trials discussed (particularly of the extreme elderly) should consider both factors together as well as separately
5. Dose is mentioned in the description of some studies but not in others, this is particularly relevant in the obese population, please be consistent in mentioning the dose used for each of the trials discussed
6. In cancer patients recent guidelines recommend DOACS for VTE prophylaxis in patients of intermediate/high risk of thrombosis, https://doi.org/10.1182/bloodadvances.2020003442, this may influence future studies
7. Line 261 In the study of Russo et al [69] among 58 patients with extreme obesity (BMI ≥40
kg/m2) and AF, 9 patients (15.5 %) showed DOAC plasma concentrations’ out of the ex- pected ranges. DOAC underdosing was found to be the only independent predictor of DOAC plasma levels out of the expected ranges (hazard ratio = 29.37; P = 0.0002) [69]. The concluding statement: According to these results, extreme obesity does not significantly impact DOAC plasma levels [69]. This statement contradicts the earlier result.
8. The conclusion is inadequate, please give a conclusion based on the evidence discussed and suggest recommendations for practice.
Author Response
- Title: The title should reflect the topic and should mention patients with AF. The current title suggests that the study includes all types of patients using DOACs.
Thanks for your suggestion. We have changed the title (in red).
- The introductory paragraph should include a description of the current guidelines to set the subsequent sections in context.
Thanks for your comment. We added a description, in the introduction, of the current guidelines.
- For each special population a summary table of the trials discussed would be useful focusing on the main outcome variables of stroke and bleeding.
Thanks for your comment. We added tables as suggested.
- Many aged patients are also frail, therefore trials discussed (particularly of the extreme elderly) should consider both factors together as well as separately
Thanks for your comment. We have described studies on frail patients, when frailty assessment was performed. Being extremely old does not necessarily mean frail.
- Dose is mentioned in the description of some studies but not in others, this is particularly relevant in the obese population, please be consistent in mentioning the dose used for each of the trials discussed.
Thanks for your comment. When not mentioned, the dosages are those indicated by the technical data sheet and the guidelines. We added a sentence to clarify this point (line 254 in red).
- In cancer patients recent guidelines recommend DOACS for VTE prophylaxis in patients of intermediate/high risk of thrombosis, https://doi.org/10.1182/bloodadvances.2020003442, this may influence future studies.
Thanks for your comment. This review is focused on DOACs in cancer patients with AF, but if necessary we can insert the reference.
- Line 261 In the study of Russo et al [69] among 58 patients with extreme obesity (BMI ≥40 kg/m2) and AF, 9 patients (15.5 %) showed DOAC plasma concentrations’ out of the ex- pected ranges. DOAC underdosing was found to be the only independent predictor of DOAC plasma levels out of the expected ranges (hazard ratio = 29.37; P = 0.0002) [69]. The concluding statement: According to these results, extreme obesity does not significantly impact DOAC plasma levels [69]. This statement contradicts the earlier result.
Thank you for your comment. The study we cited found that among extremely obese patients, not obesity but inappropriate prescription of low-dose DOAC was an independent predictor of drugs plasma levels out of the expected range. We have rephrased in the hope that the sentence will be clearer.
- The conclusion is inadequate, please give a conclusion based on the evidence discussed and suggest recommendations for practice.
Thanks for your suggestion. We changed the conclusion.
Reviewer 2 Report
In this review, the authors comprehensively reviewed the available literature on the efficacy and safety of DOACs in elderly, obese or underweight, frail, cancer patients and in patients with bioprosthetic heart valves with non-valvular atrial fibrillation (NVAF). This paper provides a very comprehensive overview of the field and is very well written.
I have only minor comments:
Suggest adding a few sentences to describe the working mechanisms of DOACs (targets, binding property etc) because the readers of this journal might not necessarily be very familiar with these drugs.
Page 4 line 202, BMI unit should be kg/m^2.
Suggest adding a graphic abstract to summarize the message.
Author Response
Suggest adding a few sentences to describe the working mechanisms of DOACs (targets, binding property etc) because the readers of this journal might not necessarily be very familiar with these drugs.
Thank you for your comment we added few sentences to describe DOACs pharmacological profile (introduction, in red).
Page 4 line 202, BMI unit should be kg/m^2.
Thank you for your comment. We modified as required.
Suggest adding a graphic abstract to summarize the message.
Thanks for your suggestion. We added a graphic abstract.
Reviewer 3 Report
The authors present a well synopsised review of the literature as well as Clinical trials with respect to DOAC use in vulnerable sub-populations. As such it is a nice benchmark review for clinicians working in this field. Carbone, Bottino and colleagues present a sound premise for the use of DOACs in a patients stratified manner. This is a relevant, topical and important concern in prescribing therapy for NVAF. The authors gave achieved the importance in highlighting this issue and noting it for future studies and identifying the need for special focus on these sub-populations.
The review is well sourced and referenced, supporting the authors hypothesis and objectives.
Points-
The review would benefit from careful editingnas there are numerous syntax, grammatical and 'non-conventional terminology' errors through-out, e.g. Ln14-17; Ln54-56; Ln69-71; Ln118-119; Ln155-157; Ln391-392.
The manuscript would benefit greatly from Tables, synopsising the various trials and outcomes.
Author Response
The review would benefit from careful editing as there are numerous syntax, grammatical and 'non-conventional terminology' errors through-out, e.g. Ln14-17; Ln54-56; Ln69-71; Ln118-119; Ln155-157; Ln391-392.
Thanks for your suggestion. We performed language editing.
The manuscript would benefit greatly from Tables, synopsising the various trials and outcomes.
Thanks for your comment. We added tables.